# Efficient Lipid Bilayer Formation by Dipping Lipid-Loaded Microperforated Sheet in Aqueous Solution

**DOI:** 10.3390/mi12010053

**Published:** 2021-01-05

**Authors:** Nobuo Misawa, Satoshi Fujii, Koki Kamiya, Toshihisa Osaki, Shoji Takeuchi

**Affiliations:** 1Artificial Cell Membrane Systems Group, Kanagawa Institute of Industrial Science and Technology, 3-2-1 Sakado, Takatsu-ku, Kawasaki, Kanagawa 213-0012, Japan; nmisawa@azabu-u.ac.jp (N.M.); sfujii0331@gmail.com (S.F.); kamiya@gunma-u.ac.jp (K.K.); tosaki@iis.u-tokyo.ac.jp (T.O.); 2Institute of Industrial Science, The University of Tokyo, 4-6-1 Komaba, Meguro-ku, Tokyo 153-8505, Japan; 3Department of Mechano-Informatics, Graduate School of Information Science and Technology, The University of Tokyo, 7-3-1 Hongo, Bunkyo-ku, Tokyo 113-8656, Japan

**Keywords:** bilayer lipid membrane, droplet interface bilayer, membrane protein, α-hemolysin, ion channel current

## Abstract

This paper describes a method for a bilayer lipid membrane (BLM) formation using a perforated sheet along with an open chamber. Microscopic observation of the formed membrane showed a typical droplet interface bilayer. We proved that the formed membrane was a BLM based on electrical measurements of the membrane protein α-hemolysin, which produces nanopores in BLMs. Unlike the conventional approach for BLM formation based on the droplet contact method, this method provides aqueous surfaces with no organic solvent coating layer. Hence, this method is suitable for producing BLMs that facilitate the direct addition of chemicals into the aqueous phase.

## 1. Introduction

Bilayer lipid membranes (BLMs) are widely used as artificial cell membranes for research on plasma membranes, membrane proteins, and various membrane-associated biomolecules [1,2,3,4]. Artificial cell membranes are applicable to platforms for the functional analysis of membrane proteins and biosensing applications [5,6,7]. Furthermore, artificial membranes integrating a nanopore membrane protein have been recently used for DNA sequencing [8] and have been studied for protein sequencing [9]. Over the years, the “painting method” [10] and “monolayer-folding method” [11] have been frequently used and regarded as simple methods for obtaining BLMs. In recent years, the “droplet contact method” [12,13,14] has received significant attention for BLM production as an easier technique that generates a droplet interface bilayer (DIB) in organic solvents containing lipid molecules.

The droplet contact method is generally carried out using aqueous droplets submerged in organic solvents. Hence, DIBs are formed in organic solvents. This configuration makes it difficult to inject some chemicals, such as drug candidates; in addition, the aqueous phase from the outside impedes the fundamental studies of membrane proteins. Similarly, for the development of odorant sensors, the oil layer on DIBs inhibit the penetration of volatile compounds, which are mostly lipophilic, into an aqueous phase. The addition of chemicals and the exchange of the isolated aqueous phase can be conducted using a pumping system or superabsorbent polymer with fluidic channels connected to a BLM formation chamber [15,16]. The system requires an accurate regulation of the solution injection and/or ejection to prevent damage to the BLMs owing to a pressure imbalance. For gas absorption into the aqueous phase, a hydrogel is applicable as a mediating material between the atmosphere and BLM [6,17]. However, the reproducible formation of a BLM at the interface between the solution and gel is a laborious process.

Herein, we propose a method of BLM formation involving a promising system for a chemical addition and gas absorption into an aqueous phase in contact with the BLMs. To establish such a BLM formation system equipped with an air–water interface, only a small amount of organic solvent along with an open chamber device is used. In this study, as a first proof of concept for BLM formation, as indicated in Figure 1, we carry out a BLM formation by dipping a perforated sheet loaded with a lipid solution into a buffer solution. We fluorinate the surface of the device for a reproducible membrane formation, and then we microscopically observe the membrane and measure the electric current in conjunction with the membrane proteins to ascertain whether the membranes are decent BLMs.

## 2. Materials and Methods

### 2.1. Materials

As the device material, we used a polymethylmethacrylate (PMMA) film and plate (i.e., Acryplen and Acrylight, purchased from Mitsubishi Chemical Co., Tokyo, Japan) for fabrication of the perforated sheets and an open chamber device, respectively. An amphiphobic coating reagent (SFCOAT, SFE-DP02H) was obtained from AGC SEIMI Chemical Co., Ltd. (Kanagawa, Japan) for surface treatment of the PMMA film.

Lipid molecules dissolved in chloroform, 1,2-dioleoyl-*sn*-glycero-3-phosphocholine (DOPC), and 1,2-dioleoyl-*sn*-glycero-3-phosphoethanolamine (DOPE) (Avanti Polar Lipids, Inc., Alabaster, AL, USA) were purchased as commercialized products, and *n*-decane (Merck KGaA, Darmstadt, Germany) was used as the solvent for the lipid solution prepared in this study. A nanopore-forming protein α-hemolysin (αHL) was used (Merck KGaA). A concentrated stock solution of αHL (20 μM) was prepared by dissolution of αHL lyophilized powder using ultrapure water from a Milli-Q system (Millipore, MA, USA). Ag/AgCl electrodes (0.4-mm diameter AG-401355, The Nilaco Co., Tokyo, Japan) were prepared in advance by dipping in a commercial bleach agent overnight. Other chemicals were obtained from Wako Pure Chemical Industries, Ltd. (Osaka, Japan) and Sigma-Aldrich Co. (St. Louis, MO, USA). All reagents were used without further purification.

### 2.2. Device Fabrication

We fabricated perforated T-shaped PMMA (Acryplen) sheets with a thickness of 75 μm and pore diameters of 500, 600, 700, and 800 μm. An open chamber device made of PMMA (Acrylight) was processed into a gourd-shaped well (3-mm depth and 24.7 mm^2^ area) with a narrow through-slit, as observed in the top view of Figure 2b. The clearance gap of the narrow slit was 1 mm. For easy observation of the formed BLMs, the inside wall of the chamber was partially planarized. The device parts (well chip, perforated sheet, and Ag/AgCl electrodes) were manually assembled using tweezers. All PMMA parts were fabricated using a milling machine (MM-100, Modia Systems Co., Ltd., Saitama, Japan) with 0.5- and 1-mm diameter end mills.

### 2.3. Surface Fluorination of PMMA and Wettability Verification

Due to concern over the disorderly spreading of the lipid solution on a T-shaped sheet and leakage of the buffer solution from a through-slit in the chamber bottom owing to the interfacial effect of lipid molecules, we employed a fluorinating agent (SFE-DP02H) for surface modification of the PMMA [18]. To estimate the efficacy of the fluorination, we verified the wettability of the fluorinated surface based on static contact angle measurements using a buffer solution and a lipid solution to describe the compositions, as described in the next subsection. We dropped 10 μL of each solution onto the PMMA (acryplen) surface before and after fluorination. The fluorination treatment was carried out by casting 20 μL of a fluorinating agent on the PMMA surface in a few square centimeters and drying for 5 min at room temperature (approx. 25 °C).

### 2.4. BLM Formation

As shown in Figure 2, two silicon rubber (AS ONE, Osaka, Japan) masks with 1-mm thickness and 3-mm diameter were applied to both sides of the perforated sheet to render the surface amphiphobic, except for the area around the through-hole, using the fluorinating agent mentioned above. Then, 0.5 μL of a lipid solution (i.e., *n*-decane containing a lipid mixture (20 mg/mL) of DOPC and DOPE at a 3:1 weight ratio) was deposited on each side of the sheet. The component of the lipid mixture was determined according to our previous study [19].

The open chamber was completely coated with the same fluorinating agent and filled with 60 μL of a buffer solution (pH 7.0; 1 M KCl, 10 mM phosphate buffer). The perforated sheet loaded with the lipid solution was manually dipped into the buffer solution in the open chamber using tweezers. Owing to the transparency of the PMMA, we could observe the through-hole of the sheet using a microscope (YDZ-3F, Yashima Optical Co., Ltd., Tokyo, Japan) from outside the chamber even when the sheet was submerged in the aqueous solution inside the chamber. In addition, we investigated the effect of fluorination of the perforated sheet on the liquid retention in the chamber and the efficiency of the BLM formation.

### 2.5. Current Recording

To electrically confirm whether the membranes formed were BLMs, we added an αHL solution with a final concentration of 1–3 nM into the chamber. The ion currents passing through the BLM using the αHL were recorded with a wired in-house developed amplifier [19] every 1 ms under a hold voltage of +60 mV. We used a simple Faraday cage made of an aluminum foil that covered the chamber device. All experiments were carried out at room temperature.

## 3. Results and Discussion

### 3.1. Efficacy of Fluorination

To verify the efficacy of the PMMA fluorination, we confirmed the contact angles of the buffer solution and the lipid solution. Fluorination increased the surface tension of the droplets in either case. The fluorination provided more water-repellent properties to the PMMA surface, which achieved repellency (Figure 3a). The contact angles changed from 70 ± 3° to 103 ± 2°. On a bare PMMA, we were unable to measure the contact angles of the lipid solution. Fluorination was drastically effective for increasing the surface tension of the lipid solution (Figure 3b). The contact angles after fluorination were 62 ± 6°. These results imply that this fluorination treatment was useful for preventing leakage of the buffer solution from a through-slit in the chamber bottom and for retaining the lipid solution around the through-hole of the T-shaped PMMA sheet. As schematically shown in Figure 2a, the T-shaped PMMA sheet was fluorinated except for the area around the through-hole, because the thinning of the lipid solution locally on the hole was favorable for efficient BLM formation.

### 3.2. Device Configuration and Preparation of BLM Formation

Figure 4a,b show the fabricated open chamber device and the perforated sheet loaded with the lipid solution. We found that the loaded lipid solution became thin on a bare PMMA and did not spread owing to the fluorinated area, as expected.

From outside the chamber, we observed the through-hole of the sheet in the chamber filled with a buffer solution, as shown in Figure 4d. Note that it is necessary to avoid an electrical short circuit between the two compartments separated by the inserted sheet for electrical measurements of the BLM systems. Therefore, we purposely designed the T-shaped sheet to penetrate the open chamber through the through-slit at the bottom to ensure sufficient insulation properties of the two compartments. As a preliminary experiment, we were able to obtain such an insulated configuration using an unperforated PMMA sheet and a chamber device filled with a buffer solution.

### 3.3. BLM Formation and Microscopic Observation

Because the PMMA surface is hydrophobic, an aqueous solution without a surfactant does not spread when applied to the narrow PMMA-enclosed spaces. However, leakage of the buffer solution from the slit at the chamber bottom is often encountered when a bare PMMA sheet and chamber are used. It seemed that the leakage was caused by a decrease in the surface tension of the buffer solution owing to the amphiphilic property of the lipid molecules. As shown in the upper image of Figure 5a, without fluorination, the buffer solution frequently exuded from the gap between the inserted sheet and the bottom of the open chamber. Simultaneously, short circuits through the through-hole frequently occurred; hence, membrane formation was not achieved. Meanwhile, no leakage of the solution was observed when a fluorinating agent was used (lower image in Figure 5a), and we obtained membranes with a probability of approximately 80% observationally, as shown in the right-hand image of Figure 5b. This result indicated that the fluorinated surface worked to prevent dripping of the lipid and buffer solutions from the bottom of the chamber. The right-hand image in Figure 5b is a representative image of the formed membrane. It shows an annulus area composed of a disordered phase of lipid molecules in the organic solvent, and a planar membrane area that appeared as a BLM. This image is consistent with some of the reported images of the DIBs [13,20,21]. Planar membrane images, such as that shown on the right side of Figure 5b, were reproducibly observed when a fluorinating agent was used for different pore diameters (500, 600, 700, and 800 μm) of the through-hole. This result supports the idea that our proposed method can be used to develop a BLM.

### 3.4. BLM Formation with αHL

We further confirmed the formation of BLM by electrically observing the incorporation of the nanopore-forming protein, αHL. As schematically shown in the upper image in Figure 6, we inserted Ag/AgCl electrodes into the open chamber and connected them to an amplifier. The lower chart in Figure 6 shows the current trace of αHL incorporated into the BLM when a perforated sheet with a 600-μm diameter through-hole was used. Based on the variation in current arising from the incorporation of αHL, the membranes formed were confirmed to be BLMs. We measured the electrical noise level for each pore diameter. The background noise levels used as the standard deviation were within the range of approximately 2–15 pA for the 500 μm, 600 μm, 700 μm, and 800 μm pore diameters. We believe that the proposed BLM formation system is suitable for measurement objects that generate a sufficiently large ion current. Although a small electric current is difficult to detect, the problem can be resolved by improving the electrode configuration and shielding performance of the chamber.

We obtained positive signals caused by αHL (1–3 nM) in BLMs with several pore diameters (500–800 μm). We also defined the criteria of positive signals as stepwise ion current changes, as shown in Figure 6. We evaluated the stability of BLMs in our system based on the time between the first insertion of αHL and the rupture of BLM. The results were 7.9 ± 5.1, 4.8 ± 2.4, 1.7 ± 1.2, and 1.4 ± 1.1 min (N = 3, mean ± standard deviation) for the 500, 600, 700, and 800 μm pore diameters, respectively, indicating that the smaller pore diameter would be favorable for BLMs stability. The αHL reconstitution probability with a pore diameter of 600 μm was 65% (=30/46). We found that appropriate BLMs for αHL were formed in this system.

Unlike a conventional method for producing DIBs, only a small amount of organic solvent, 1.0 μL of *n*-decane, was used with our proposed method. Therefore, it is expected that the produced solution surface will not be covered with an organic solvent layer. In case that entire *n*-decane solution (1 μL) spreads in the top surface of the chamber (24.7 mm^2^), the expected thickness of the *n*-decane layer on the buffer solution is about 40 μm at a maximum, which would quickly disappear due to its volatility. In the actual experiment, we observed that the *n*-decane droplet mostly remained on the T-shaped perforated sheet due to the fluorination when we dipped the sheet in the buffer solution. Therefore, even if the *n*-decane layer formed on the surface of the buffer solution, it would soon volatilize and eventually an air-water interface would appear. Because the exposed air–water interface will help the penetration of volatile compounds from outside into the aqueous phase, our proposed system is potentially adequate for odorant sensing with olfactory receptors reconstructed in a BLM system [22]. Although the spontaneous evaporation of an aqueous solution is a concern, the integration of our system with water supply installations and the use of a water-resistant breathable fabric [23] will provide a potential solution.

## 4. Conclusions

We demonstrated a BLM formation by dipping an acrylic perforated sheet partially coated with a lipid solution into an aqueous solution in an open chamber. For a highly reproducible BLM formation, the perforated sheet and open chamber required a surface fluorination. The pore diameter of the perforated sheet ranged from 500 to 800 μm (thickness of 75 μm) and met the requirement for BLM formation. The authenticity of the BLM formation was validated through a nanopore formation with αHL. Owing to the use of a small amount of organic solvent in the BLM formation method, it is expected that the surfaces of the aqueous solutions will become exposed to ambient air. Hence, we consider this configuration of the BLM system to be suitable for additional chemical injection from the outside and the absorption of airborne molecules from the atmosphere into the liquid phase.

## Figures and Tables

**Figure 1 micromachines-12-00053-f001:**
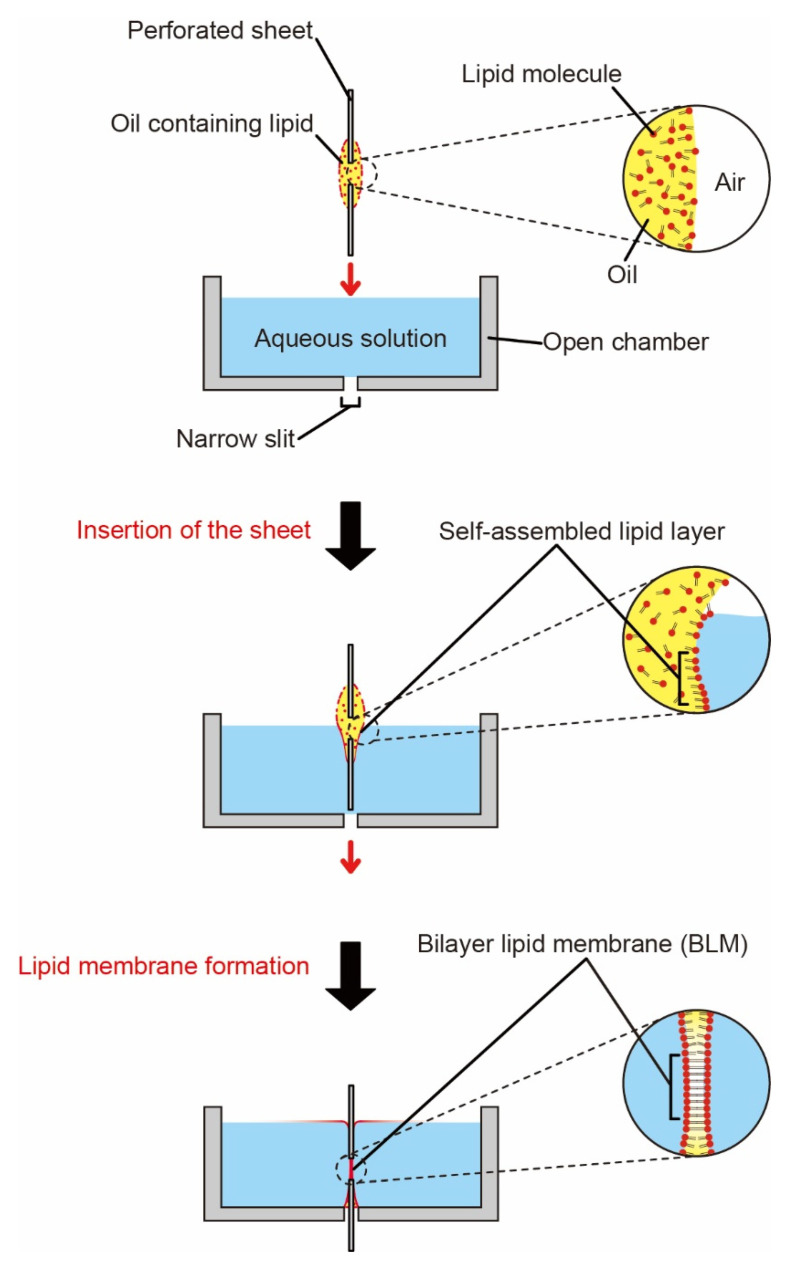
Schematic sectional images of bilayer lipid membrane (BLM) formation by dipping a perforated sheet coated with a lipid solution into an aqueous solution. A through-slit in the chamber bottom allows penetration of the sheet. BLM is spontaneously formed at the interface of the two isolated compartments.

**Figure 2 micromachines-12-00053-f002:**
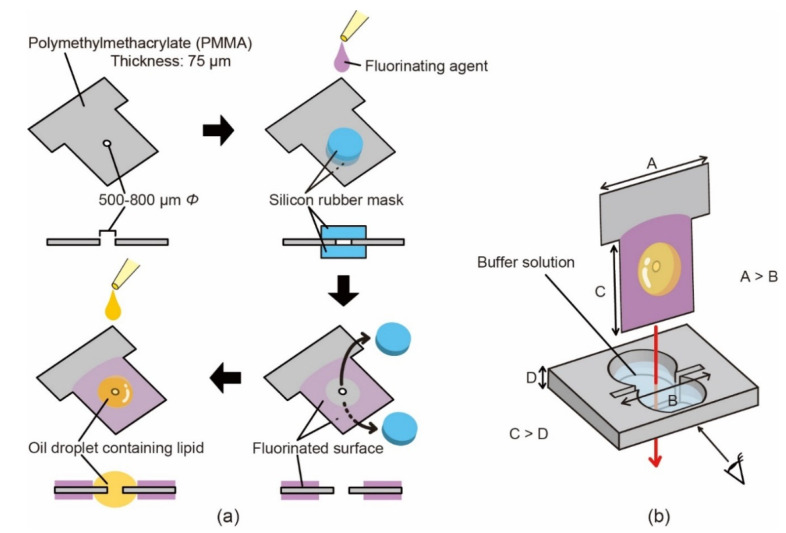
Schematic illustration of (**a**) the stepwise procedure of perforated sheet modification and (**b**) perforated sheet insertion into an open chamber device used for BLM formation.

**Figure 3 micromachines-12-00053-f003:**
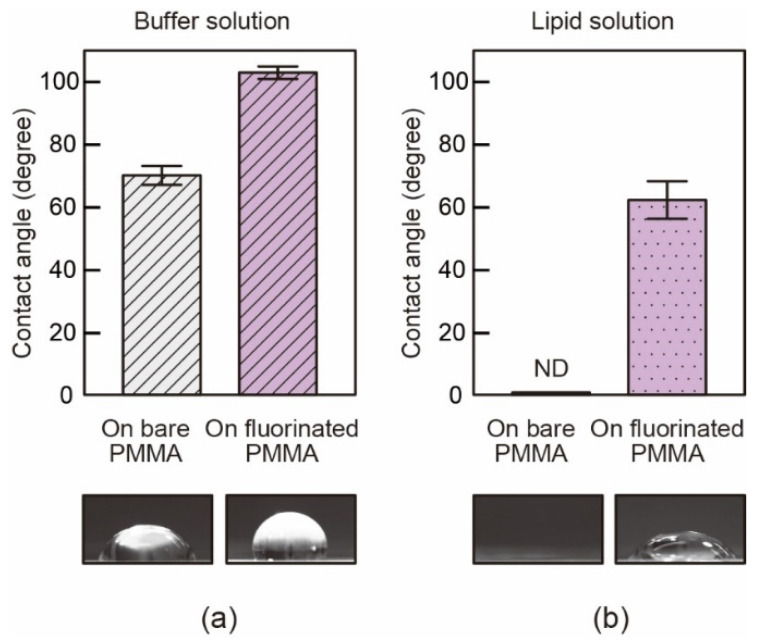
Results of static contact angle measurements for (**a**) buffer solution and (**b**) lipid solution on PMMA surface before and after fluorination. Error bar: ±standard deviation. Each data point was obtained by five experimental measurements. Representative microscopic images of the 10-μL droplet are the lower images.

**Figure 4 micromachines-12-00053-f004:**
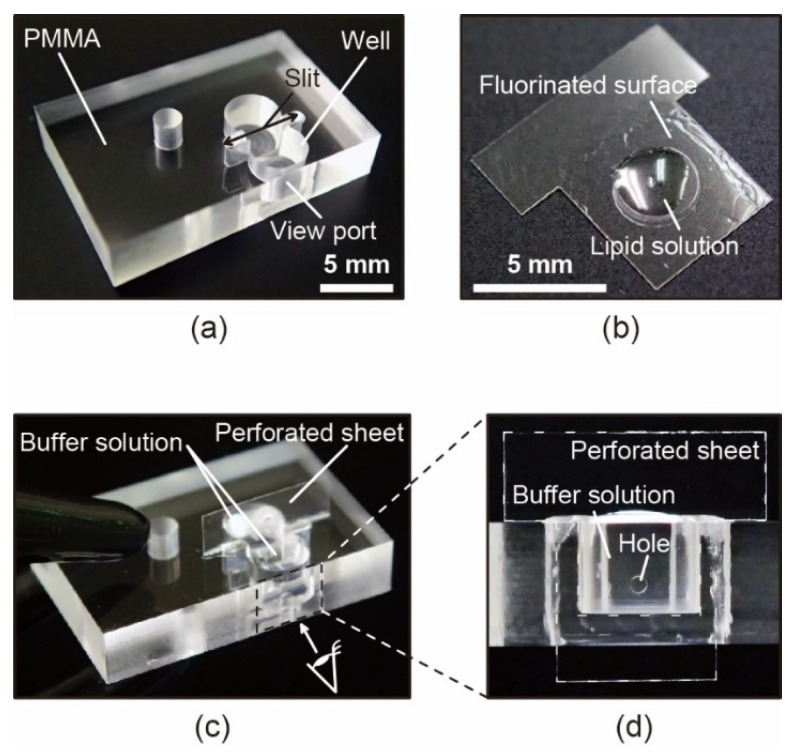
(**a**) Image of the open chamber device made of polymethylmethacrylate (PMMA). (**b**) T-shaped perforated sheet with a droplet of lipid solution on the surface. (**c**) Image of the open chamber device containing the perforated sheet. (**d**) Front view of the perforated sheet in the open-chamber device.

**Figure 5 micromachines-12-00053-f005:**
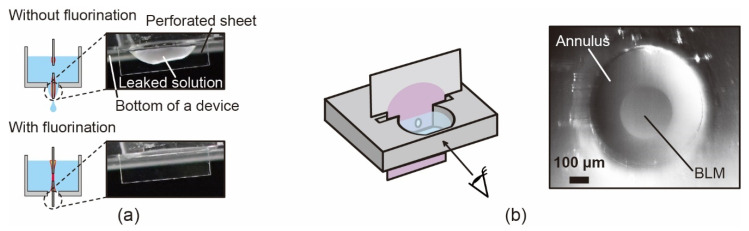
(**a**) Penetration of separator causing (**upper**) leakage and (**lower**) no leakage of solution from the slit of the open chamber. (**b**) (**Left**) Observation angle of the BLM formed in the open chamber and (**right**) image of the BLM.

**Figure 6 micromachines-12-00053-f006:**
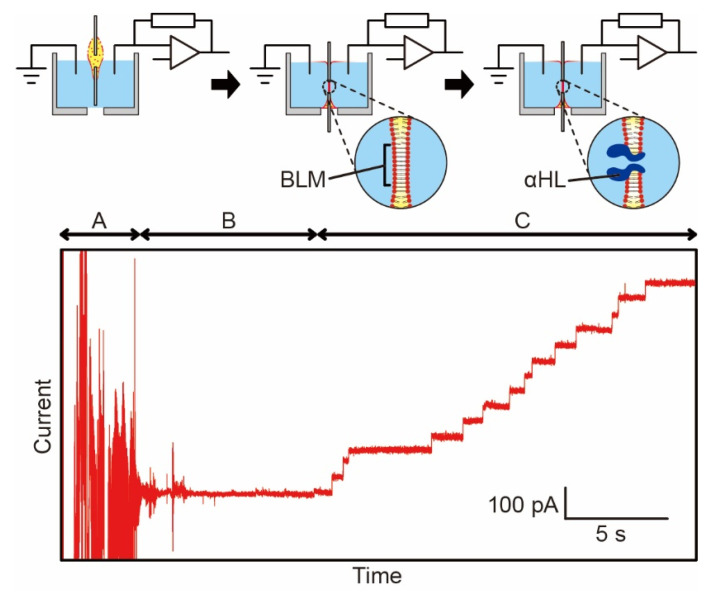
(**Upper**) Schematic procedure of “perforated sheet insertion” to “intact BLM formation and ion channel current arising from reincorporation of αHL into the BLM”. (**Lower**) Variation in ion channel current caused by αHL in BLM. (**A**) Initial noise generated by sheet insertion. (**B**) Intact BLM. (**C**) Current changes caused by αHL.

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
