# Peer review of "Efficient Lipid Bilayer Formation by Dipping Lipid-Loaded Microperforated Sheet in Aqueous Solution"

_micromachines, 2021, doi:10.3390/mi12010053_

Round 1

Reviewer 1 Report

A method for the preparation of bilayer lipid membrane using a perforated sheet is presented. Methods and results are presented well. I recommend to accept the article with a minor revision. #Comment Provide the details of α-hemolysin sample preparation for the experiment.

Author Response

We really thank for your kind comment. Concerning the preparation of αHL sample, we added a following sentence in Materials and Methods section.

Line 75-77.

A concentrated stock solution of αHL (20 μM) was prepared by dissolution of αHL lyophilized powder using ultrapure water from a Milli-Q system (Millipore, MA, USA).

Reviewer 2 Report

This paper describes the lipid bilayer formation method that a perforated sheet loaded with a lipid solution is inserted into an aqueous bath solution. This proposed method may allow to add chemicals and adsorb gas into the aqueous solution because an oil layer is not formed on the aqueous solution. However, the authors did not discuss the formation of the oil layer and the availability of the chemical injection and gas adsorption. In addition, discussions of the BLM formation and recorded αHL channel currents are very poor. There are several concerting points as listed below.

  1. The recorded channel currents of αHL (Figure 6) are not correct. The authors described the experimental conditions of channel current measurements as below. Buffer solution: (pH 7.6; 5 mM 4-(2-hydroxyethyl)-1-piperazineethanesulfonic acid, 96 mM NaCl, 2 mM KCl, 5 mM MgCl2, and 0.8 mM CaCl2. Applied potential: + 60 mV.

However, the channel current is around 60 pA, which is too high.

The authors should discuss the channel currents and prove that the channel currents are from αHL nanopores.

  1. The authors should evaluate the stability of BLMs. In this method, the hole diameter of the perforated-sheet and the area of the BLM are much larger, and the volume of the oil is much lower as compared with the previously reported droplet contact methods. Because of these conditions, the stability of the BLMs would be lower than that of conventional DIBs.

  1. The authors mentioned that “Unlike a conventional method for producing DIBs, only a small amount of organic solvent, 1.0 μL of n-decane, was used with our proposed method. Therefore, it is expected that the produced solution surface will not be covered with an organic solvent layer.”. However, it is the extremely poor discussion of the formation of the oil layer on the droplets. The authors should discuss the formation of the organic solvent layer in more detail.

Author Response

1.

The recorded channel currents of αHL (Figure 6) are not correct. The authors described the experimental conditions of channel current measurements as below. Buffer solution: (pH 7.6; 5 mM 4-(2-hydroxyethyl)-1-piperazineethanesulfonic acid, 96 mM NaCl, 2 mM KCl, 5 mM MgCl2, and 0.8 mM CaCl2. Applied potential: + 60 mV. However, the channel current is around 60 pA, which is too high. The authors should discuss the channel currents and prove that the channel currents are from αHL nanopores.

We deeply appreciate your careful confirmation for our description of the sample preparation. There was a misdescription about compositions of the buffer solution. The correct composition of the buffer solution is “pH 7.0; 1 M KCl, 10 mM phosphate buffer”. Accordingly, the applied voltage and the channel currents are correct. Hence, we revised the corresponding part (Line 108) for the above-mentioned compositions. We are truly thankful for your comment again.

2.

The authors should evaluate the stability of BLMs. In this method, the hole diameter of the perforated-sheet and the area of the BLM are much larger, and the volume of the oil is much lower as compared with the previously reported droplet contact methods. Because of these conditions, the stability of the BLMs would be lower than that of conventional DIBs.

Thank you very much for your fruitful comment. As the reviewer pointed out, we confirmed that BLM became stable with decrease of the hole diameter. We added the sentences as indicated below.

Line 202-205.

We evaluated the stability of BLMs in our system based on the time between the first insertion of αHL and the rupture of BLM. The results were 7.9 ± 5.1, 4.8 ± 2.4, 1.7 ± 1.2, and 1.4 ± 1.1 min (N=3, mean ± standard deviation) for the 500, 600, 700, and 800 μm pore diameters, respectively, indicating that the smaller pore diameter would be favorable for BLMs stability.

3.

The authors mentioned that “Unlike a conventional method for producing DIBs, only a small amount of organic solvent, 1.0 μL of n-decane, was used with our proposed method. Therefore, it is expected that the produced solution surface will not be covered with an organic solvent layer.”. However, it is the extremely poor discussion of the formation of the oil layer on the droplets. The authors should discuss the formation of the organic solvent layer in more detail.

We truly feel grateful for your instructive advice. When we dipped the oil (n-decane)-loaded T-shaped perforated sheet in buffer solution, oil droplet mostly remained on the sheet due to the fluorination, and thus less than 1 μl n-decane spread air-water interface of the buffer solution in the chamber. If all n-decane of 1 μl would spread in the top surface of the chamber (24.7 mm2), the expected thickness of n-decane layer is about 40 μm. Since n-decane is volatile at room temperature, it is considered that the device configuration provides air-water interface. Hence, we added sentences for the discussion as follows.

Line 210-216.

In case that entire n-decane solution (1 μl) spreads in the top surface of the chamber (24.7 mm2), the expected thickness of the n-decane layer on the buffer solution is about 40 μm at a maximum, which would quickly disappear due to its volatility. In the actual experiment, we observed that the n-decane droplet mostly remained on the T-shaped perforated sheet due to the fluorination when we dipped the sheet in the buffer solution. Therefore, even if the n-decane layer is formed on the surface of the buffer solution, it would soon volatilize and eventually an air-water interface would appear.

Round 2

Reviewer 2 Report

The authors completely modified the revisions.